# *Arabidopsis* COP1 guides stomatal response in guard cells through pH regulation

Seoyeon Cha [1,3], Wang Ki Min[1,3] & Hak Soo Seo [1,2✉]

Plants rely on precise regulation of their stomatal pores to effectively carry out photosynthesis while managing water status. The *Arabidopsis* CONSTITUTIVE PHOTO-MORPHOGENIC 1 (COP1), a critical light signaling repressor, is known to repress stomatal opening, but the exact cellular mechanisms remain unknown. Here, we show that COP1 regulates stomatal movement by controlling the pH levels in guard cells. *cop1-4* mutants have larger stomatal apertures and disrupted pH dynamics within guard cells, characterized by increased vacuolar and cytosolic pH and reduced apoplastic pH, leading to abnormal stomatal responses. The altered pH profiles are attributed to the increased plasma membrane (PM) $H^+$-ATPase activity of *cop1-4* mutants. Moreover, *cop1-4* mutants resist to growth defect caused by alkali stress posed on roots. Overall, our study highlights the crucial role of COP1 in maintaining pH homeostasis of guard cells by regulating PM $H^+$-ATPase activity, and demonstrates how proton movement affects stomatal movement and plant growth.

[1] Department of Agriculture, Forestry and Bioresources, Research Institute of Agriculture and Life Sciences, Seoul National University, Seoul 08826, Republic of Korea. [2] Bio-MAX Institute, Seoul National University, Seoul 08826, Republic of Korea. [3] These authors contributed equally: Seoyeon Cha, Wang Ki Min. ✉email: seohs@snu.ac.kr

Stomatal pores in leaves are surrounded by two guard cells and regulate gaseous exchange and water transpiration in plants. While the opening and closure of stomata are triggered by environmental factors such as light, temperature, and $CO_2$ concentration[1,2], their physical changes involve ionic movements, including $K^+$, $Cl^-$, and $NO_3^-$[3–5]. These cations and anions constitute the buffer system inside the cell and participate in pH regulation alongside $H^+$ exchanges, underscoring the significance of guard cell pH in stomatal movements[6]. To enhance photosynthesis and water use efficiency, a deeper understanding of guard cell pH regulation is crucial. Therefore, this study focuses on investigating the role of Arabidopsis CONSTITUTIVE PHOTOMORPHOGENIC 1 (COP1) in guard cell pH regulation, as it has been previously linked to regulating stomatal aperture and density for stomatal conductance[7–9].

COP1 is a highly conserved E3 ubiquitin ligase in eukaryotes to ubiquitinate various protein substrates to consequently mark them for degradation[10]. In plants, COP1 serves as a central repressor of photomorphogenesis[11–13], and also regulates multiple pathways such as flowering time[14], hormone signaling[15], and circadian rhythm[16]. Particularly, in guard cells, COP1 negatively influences stomatal opening under the light and ABA signaling so that cop1 mutants not only exhibit constitutively large and open stomata, also are insensitive to dark- and ABA-induced stomatal closure[7,17]. Regarding the light signaling, COP1 acts downstream of CRY and PHOT to inhibit blue light-induced stomatal opening[7] and also suppresses phyA/B-mediated red and far-red light-induced stomatal opening[18], while stimulating stomatal closure under UV-B light perception[19,20]. Additionally, COP1 promotes the ABA signaling pathway through the ubiquitination of PP2C family proteins and microtubule destabilization[17,21]. This COP1-mediated degradation of PP2C leads to the release of the kinase OPEN STOMATA 1 (OST1), which activates Slow Anion Channels (SLAC1) and Quick Anion Channels (QUAC), whereas inactivating $K^+$-inward rectifying channel (KAT1), leading to anion efflux and stomatal closure[22–25]. Indeed, cop1 mutants display reduced SLAC1 channel activity, but no change in $K^+$ channel activity[21].

Despite the well-established role of COP1 in stomatal regulation, the underlying cellular mechanisms by which it affects guard cells in vivo that make cop1 mutants so recalcitrant in responding to any environmental and endogenous signals to adjust the size of stomata are still an open question. It is conceivable to think of a common landscape that can embrace the light and ABA signaling in guard cells. Arabidopsis plasma membrane (PM) $H^+$-ATPases, which pump out protons from the cytosol, are widely regarded as the final signaling destination for various pathways, such as in light, auxin, and immunity[26–30]. We thus here posit that pH serves as an important factor in stomatal guard cells generally applied for various stomatal responses. It is noteworthy that dominant mutants of ARABIDOPSIS $H^+$-ATPASE 1 (AHA1), ost2-2D, which have constitutive activation of the PM $H^+$-ATPase to open the stomata, are not responsive to ABA for the closure[31]. This insensitivity of ost2-2D mutants was recently corroborated by the detection of a higher cytosolic pH in their guard cells[32], building on earlier research in 1992 that cytosolic pH change precedes stomatal closure[33]. Cytosolic alkalinization has been implicated in both ABA-induced stomatal closure and fusicoccin (FC)-induced stomatal opening, making it difficult to uncover its homeostatic mechanism[32,34,35]. Furthermore, vacuolar pH significantly affects stomatal regulation, with $K^+$, $Na^+/H^+$ exchangers (NHX1/2) needed for alkalinization during opening[36] and V-PPase and V-ATPase necessary for acidification during closure[37–39].

In this study, we aim to reassess how COP1 modulates stomatal movement in guard cells, with a particular emphasis on cellular pH as a key biochemical factor for stomatal regulation. We investigate the altered pH dynamics in cop1 mutants and examine how different pH levels affect stomatal physiology and plant growth, highlighting the importance of guard cell pH in integrating various signals to determine whether stomata should be open or closed.

## Results

**cop1 mutants exhibit distinct vacuolar dynamics in guard cells.** cop1 mutants have abnormally large stomatal apertures, causing a cooling effect on their leaf temperature due to the increased water transpiration[17]. Notwithstanding exposure to 72 h of darkness, ABA, flg22, and other abiotic stresses, the guard cells remain turgid[40], implying a potential issue with vacuolar dynamics, which directs stomatal movement through the turgor-driven mechanism. We therefore carried out examinations of their vacuolar properties in guard cells. Using the GFP-linked vacuolar $H^+$-pyrophosphatase (V-PPase) as a tonoplast marker[41], we first examined the vacuolar morphology of cop1 mutants. Figure 1a shows the single and enlarged vacuoles of cop1-4 mutants that occupied almost 80% of a single cell and represented a 1.4-fold increase in volume, compared to the several, separate vacuoles seen in WT, conferring larger stomatal apertures (Fig. 1b–d). This enlarged vacuolar morphology of cop1-4 mutants was consistent under ABA treatment (Fig. 1a). Given the deviation of cop1 mutants from the normal vacuolar volume range of wild-type, we surmised that the vacuolar pH in guard cells may also be altered.

To measure the vacuolar pH, we utilized the vacuole-specific loading dye, 2′,7′-bis-(2-carboxyethyl)-5-(and-6)-carboxyfluorescein (BCECF), which changes its excitation peak based on pH[36,42]. It exhibits maximum excitation spectra at 458 nm for acidic values and 488 nm for alkaline values. Therefore, we calculated the luminal pH of vacuoles ratiometrically expressed as 488/458 and converted it into the absolute pH values through a pH titration curve generated with guard cells. We incubated the epidermal peels in a series of buffers ranging from 4.3 to 7.8 (Supplementary Fig. 1). It is intriguingly noted that guard cells incubated in acidic buffers closed stomata and shrank, whereas those incubated in alkaline conditions opened stomata and swelled. These findings suggest that pH may have a causal impact on stomatal dynamics, with alkaline conditions leading to larger stomata. Further, we noted an unexpected nature of vacuolar pH during the development of guard cells[43]. Young guard cells had acidic vacuoles that became alkaline as they matured (Supplementary Fig. 2). This observation supports the notion that V-PPase-mediated acidification is highly expressed in budding cells, promoting vacuolar and cellular expansion[44,45]. Given these observations, we focused on mature guard cells with a cell length greater than 18 μm for subsequent analyses to rule out any developmental-dependent pH discrepancies.

We observed that cop1-4 mutants have larger stomata with vacuoles that maintain an alkaline pH of 7.0, compared to WT with pH 6.8 (Fig. 1e, f). We further investigated the response of these vacuoles to ABA or dark treatments, and found that they did not undergo the expected vacuolar acidification required for stomatal closure[37]. In contrast, WT vacuoles showed a decrease in pH of about 0.4–0.8 units in response to these treatments, while cop1-4 only exhibited minor changes of about 0.1–0.3 units (Fig. 1e, f). This outcome suggests a deficiency in proton import into vacuoles in cop1 mutants, which may explain their resistance to stomatal closure. The observed constitutively open and larger stomata in cop1-4 mutants may be attributed to the higher pH of the single and occupied vacuoles. These distinct vacuolar properties of cop1 mutants suggest the underlying signaling barriers that either inhibit or strengthen the regular pathway.

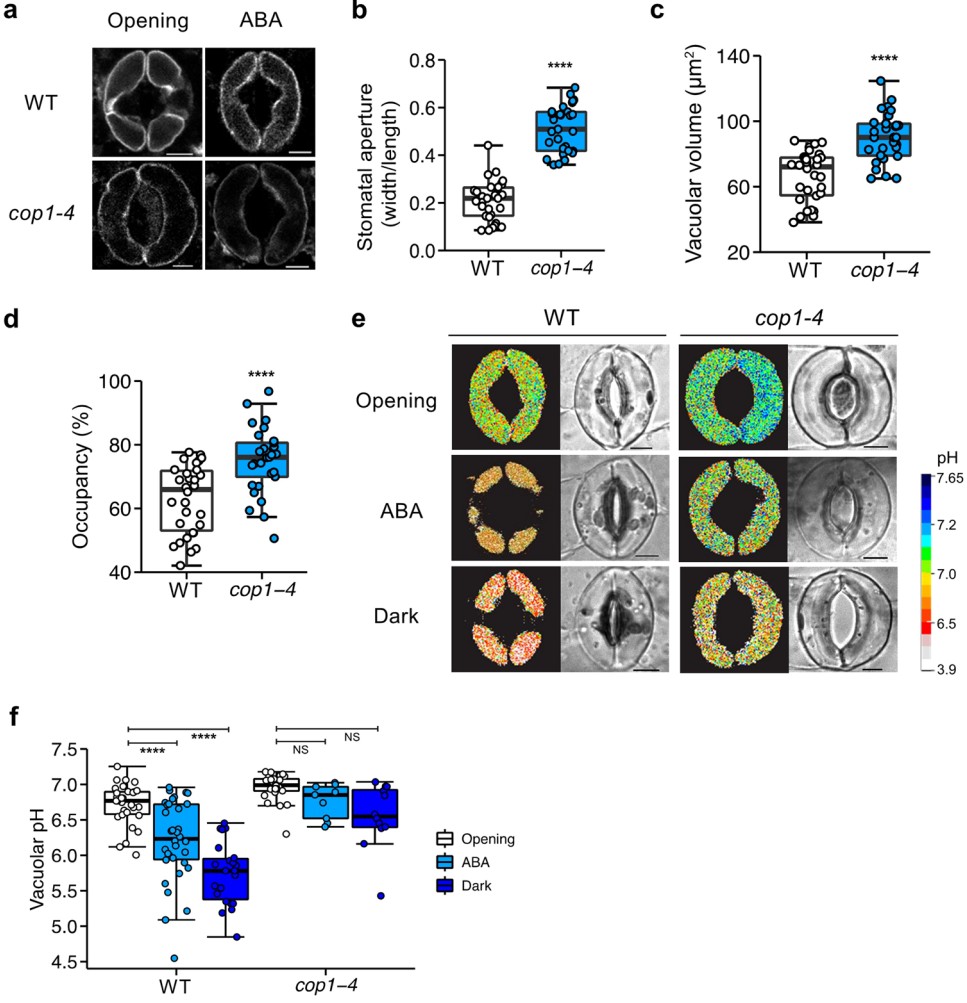

**Fig. 1 cop1-4 mutants exhibit altered vacuolar properties in guard cells. a** Vacuolar structures in guard cells of wild-type (WT) and *cop1-4* mutants visualized with the tonoplast marker VACUOLAR H+-PYROPHOSPHATASE 1 (VHP1)-GFP after 2 h of light-induced stomatal opening, followed by treatment with or without 10 µM abscisic acid (ABA) for an hour. (Scale bar, 5 µm.) Quantitative analysis of three guard cell parameters after stomatal opening. Stomatal aperture index is calculated by dividing the stomatal width by its length (**b**), combined vacuolar volume (**c**), and vacuolar occupancy (**d**). Note that these were obtained from the same guard cells used in (**f**) for stomatal opening. ****$P < 0.0001$; two-sample $t$ test; $n = 30$ (WT) and $n = 29$ (*cop1-4*). Boxplots show the median (center line) and the 75th and 25th percentiles (edges of the box) of the data; whiskers extend to 1.5 times the interquartile range. **e** Representative pseudocolor ratiometric images of vacuolar pH in guard cells of wild-type (WT) and *cop1-4* mutants were obtained after 2 h of light-induced stomatal opening, followed by treatment with or without 10 µM ABA for 1 h, or 1 h of darkness. Guard cells were loaded with the pH-sensitive vacuolar loading dye 2′,7′-bis-(2-carboxyethyl)-5-(and-6)-carboxyfluorescein (BCECF-AM), and the images were generated by dividing the emission images acquired in the 488 nm channel by those obtained in the 458 nm channel. Ratiometric images (left) and bright-field (right). (Scale bar, 5 µm.) **f** Mean vacuolar pH values for entire vacuoles in guard cells of (**e**). ****$P < 0.0001$ and NS, not significant ($P > 0.05$); two-way ANOVA; Tukey's HSD; $n = 9–34$.

**Altered guard cell pH dynamics in *cop1* mutants.** To gain a better understanding of the modified pH within vacuoles, we next profiled cytosolic pH through transgenic plants expressing ClopHensor, a genetically-encoded pH sensor[35], under various conditions modifying stomatal aperture (ABA, dark, and FC). This enabled us to monitor cytosolic pH in vivo at the single-cell level using a similar ratiometric calculation of the 488/458 ratio as that previously employed on BCECF. Together with in vivo calibration data (Supplementary Fig. 3), Fig. 2 illustrates that the basal levels of cytosolic pH were higher in *cop1-4* guard cells than in WT, as reflected by stronger signals detected at the 488 nm within the guard cell border. This result bolsters the link between cytosolic pH and stomatal aperture and is consistent with previous research demonstrating that *ost2-2D* mutants with constitutive stomatal opening have alkalinized cytosol in guard cells[32]. Elevated intracellular pH levels in guard cells of *cop1-4*

mutants indicate a homeostatic mechanism in which pH changes in the vacuole and cytosol occur in the same direction[46]. After 10 min of ABA treatment, cytosolic pH of both WT and *cop1-4* mutants slightly increased (Fig. 2), while 1 h of ABA treatment to fully induce stomatal closure of WT resulted in cytosolic acidification in both plants (Supplementary Fig. 4). Dark treatment significantly reduced cytosolic pH of WT and *cop1-4* mutants, but still *cop1-4* mutants showed higher cytosolic pH than WT. Lastly, FC treatment induced cytosolic alkalinization for both WT and *cop1-4* mutants.

We then evaluated the apoplastic pH using the membrane-impermeable pH indicator 8-hydroxypyrene-1,3,6-trisulfonic acid trisodium salt (HPTS), whose pH is determined ratiometrically by 488/405 nm[47]. A leaf disc was mounted in a stomatal opening buffer with HPTS added and immediately used for imaging. Combined with in situ HPTS calibration data (Supplementary

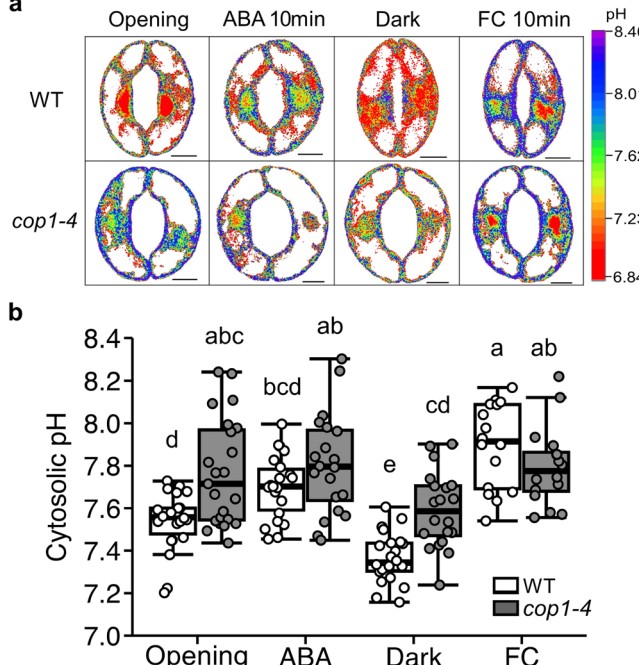

**Fig. 2 Cytosolic pH of *cop1-4* mutants are more alkaline than WT.**
**a** Representative pseudocolor ratiometric images of guard cells of wild-type (WT) and *cop1-4* mutants each expressing ClopHensor. Detached leaves were incubated with stomatal opening buffer for 2 h, and then treated with 10 μM abscisic acid (ABA) for 10 min or with dark for 1 h. To assess the impact of fusicoccin (FC), dark-treated leaves were incubated with 10 μM fusicoccin for 1 h. Fluorescence intensity of cytosolic pH in guard cells of WT and *cop1-4* mutants was measured directly from detached leaves using confocal laser scanning microscope (Leica SP8 X). Pseudocolor ratiometric images were generated by dividing the emission images obtained at 488 nm by those acquired at 458 nm. (Scale bar, 5 μm.) **b** Quantification of ratiometric values for cytosolic pH in guard cells shown in (**a**). Different letters indicate statistically significant differences between the plants ($P < 0.05$); one-way ANOVA; Tukey's HSD.

Fig. 5), our results demonstrated that the apoplastic pH in *cop1-4* mutants was significantly lower than in WT, with an average difference of 0.13 units (Fig. 3a, c). This acidic apoplast, in contrast to the alkaline intracellular environment, creates a higher pH gradient across the guard cell in *cop1-4* mutants. This increased pH gradient may contribute to the constitutively open stomata phenotype in *cop1-4* mutants by affecting ion fluxes across the plasma membrane.

Given the acidic apoplastic space observed in *cop1-4* mutants, we investigated whether they follow the typical ABA signaling pathway characterized by a transient apoplastic alkalinization[48]. Short-term apoplastic pH increase is known to be involved in the plant stress response, such as to salinity, drought, and pathogen attack, and serves to initiate stomatal closure[49]. Interestingly, it turned out that ABA-induced a prominent transient apoplastic alkalinization in WT, resulting in a 0.13 pH unit increase, while the increase in apoplastic pH was minimal in *cop1-4* mutants (Fig. 3b, c). This dampened response to ABA may block or lower the extent of the normal progression of signaling required to stimulate stomatal closure in *cop1-4* mutants. Overall, the altered basal pH levels in *cop1-4* mutant guard cells demonstrate the role of COP1 in regulating pH and help to explain their insensitivity to various factors in the stomatal response.

**COP1 acts negatively on proton pumping activity.** We then sought to determine the underlying causes of the altered pH profiles in *cop1* mutants. We first analyzed the transcript levels of PM H$^+$-ATPases, including *AHA1* and *AHA2*, but detected no significant changes (Supplementary Fig. 6). As the next possibility, we considered post-translational modifications and indirectly assayed proton pumping activity by incubating seedlings in MS media supplemented with bromocresol purple, a pH indicator dye that changes from purple to yellow as it becomes acidic. Our results showed enhanced proton extrusion in *cop1-4* mutant seedlings compared to WT, as manifested by the clear yellow color over the roots of mutants (Fig. 4a and Supplementary Fig. 7). Rhizosphere acidification was inhibited with Na$_3$VO$_4$ treatment in both WT and *cop1-4* mutants, indicating that PM H$^+$-ATPase activity caused the proton extrusion in root cells

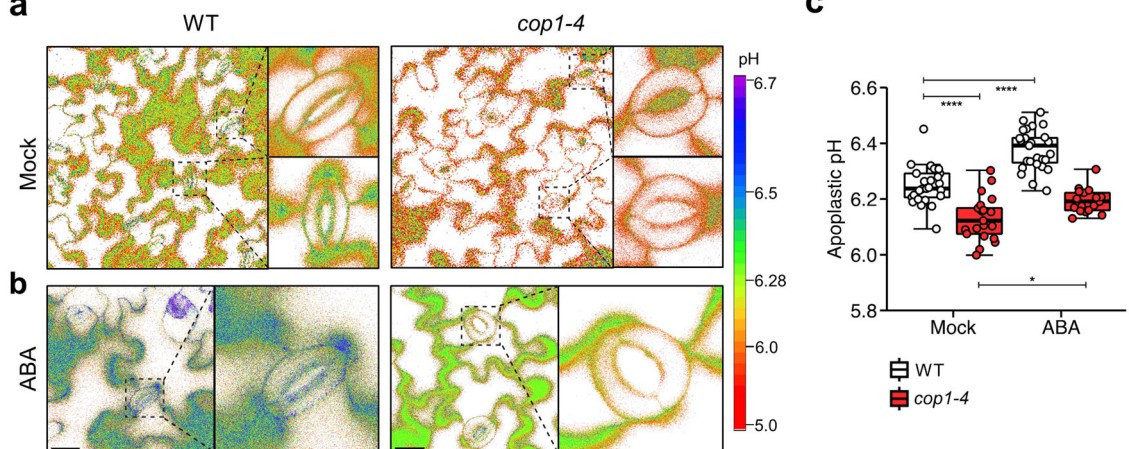

**Fig. 3 *cop1-4* mutants exhibit constitutively acidic apoplast in leaf epidermal cells. a, b** Representative pseudocolor images of apoplastic pH in the epidermis of wild-type (WT) and *cop1-4* mutants visualized using 8-hydroxypyrene-1,3,6-trisulfonic acid trisodium salt (HPTS) staining with or without abscisic acid (ABA) in the stomatal opening buffer. Staining and observation were conducted simultaneously and instantly. Ratiometric images were generated by dividing the emission images obtained in the 488 nm channel by those acquired in the 405 nm channel. Calculated values were converted into absolute pH values using the in situ HPTS calibration curve (Supplementary Fig. 3). Right panels show expansions of the left panels, respectively. (Scale bar, 20 μm.) **c** Mean apoplastic pH values for guard cells shown in (**a, b**). ****$P < 0.0001$ and *$P < 0.05$; two-way ANOVA; Tukey's HSD; $n = 17$ to 27. Boxplots show the median (center line) and the 75th and 25th percentiles (edges of the box) of the data; whiskers extend to 1.5 times the interquartile range.

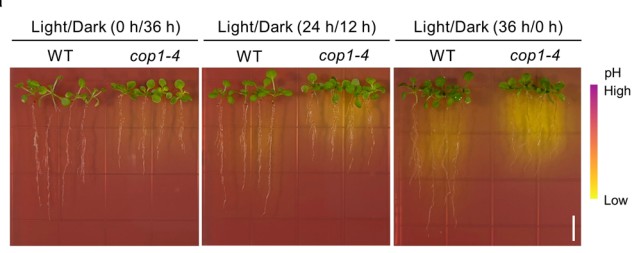

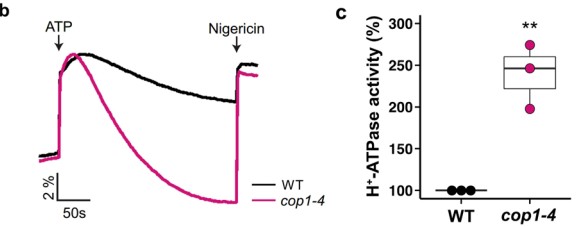

**Fig. 4 COP1 negatively regulates proton pumping activities.**
**a** Rhizosphere acidification assays were conducted using wild-type (WT) and *cop1-4* mutants. Eleven-day-old seedlings were transferred to ½ MS media containing 0.003% of pH indicator dye bromocresol purple and grown vertically. Seedlings were then subjected to dark, long-day, and continuous light conditions, respectively, as indicated, and color changes were recorded after 36 h. (Scale bar, 1 cm.) The experiment was conducted three times, and one representative experiment is shown in (**a**). **b** Plasma membrane H$^+$-ATPase activity was measured using WT and *cop1-4* mutants. Plasma membrane vesicles were isolated from the leaves of WT and *cop1-4* mutant plants, and treated with reaction buffer containing pH-sensitive fluorescence probe quinacrine. H$^+$-ATPase activation was induced by adding 3 mM ATP, and pH decrease inside the vesicles was measured using spectrofluorometer. Proton gradient inside the vesicles was collapsed by adding uncoupler nigericin. The experiment was conducted three times, and one representative experiment is shown in (**b**). **c** Mean H$^+$-ATPase activity of WT and *cop1-4* mutants of (**b**). H$^+$-ATPase activity of WT was set to 100%, and the relative H$^+$-ATPase activity of *cop1-4* mutants was shown as percentage of the activity of WT. **P < 0.01; two-sample t-test; Boxplots show the median (center line) and the 75th and 25th percentiles (edges of the box) of the data; whiskers extend to 1.5 times the interquartile range.

(Supplementary Fig. 7). Moreover, both WT and *cop1-4* seedlings showed slightly lower proton extrusion level in MS media supplemented with ABA (Supplementary Fig. 7; upper panel) compared to control media, which is in line with other data (Fig. 3).

To further explore the role of COP1 in rhizosphere acidification, we questioned whether its effect on proton extrusion was dependent on light. We revealed that the degree of rhizosphere acidification was light-dependent in both *cop1-4* mutants and WT, gradually increasing as seedlings transitioned from complete darkness to continuous light over 36 h (Fig. 4a). This suggests that the PHOT-mediated blue light signaling pathway to activate PM H$^+$-ATPases is not completely blocked by COP1, despite its known downstream function[7]. In light of *cop1-4* displaying increased acidification even under continuous light, we rather concluded that COP1 quantitatively inhibits the blue light signaling pathway by controlling the stability of downstream components[50].

Based on rhizosphere acidification assay, we speculated that *cop1-4* mutants may have enhanced proton pumping activity also in the shoot part, where the guard cells reside. To directly measure PM H$^+$-ATPase activity of *cop1-4* mutants, we isolated PM vesicles from the rosette leaves of WT and *cop1-4* mutants with aqueous two-phase separation method[51]. Each PM vesicles

harboring same amount of PM H$^+$-ATPase protein (Supplementary Fig. 8) was subjected to PM H$^+$-ATPase activity measurement using pH-sensitive fluorescence probe quinacrine[51,52]. The results clearly demonstrated that *cop1-4* mutants have about 2.5-fold higher PM H$^+$-ATPase activity than WT (Fig. 4b, c). Together with the rhizosphere acidification assay, these data strongly indicate that COP1 negatively regulates proton pumping activity.

**Enhanced PM H$^+$-ATPase activity contributes to stomatal opening and alkaline tolerance of *cop1-4* mutants.** As *cop1-4* mutants had higher PM H$^+$-ATPase activity than WT, we asked whether PM H$^+$-ATPase activity is the key factor for constitutive stomatal opening phenotype of *cop1-4* mutants. To investigate this, we treated guard cells of WT and *cop1-4* mutants with sodium orthovanadate (Na$_3$VO$_4$), a PM H$^+$-ATPase inhibitor, and compared the stomatal aperture with ABA, dark, and FC-treated guard cells. It was observed that Na$_3$VO$_4$-treated *cop1-4* mutants showed reduced stomatal aperture to the similar extent of light-treated WT, just as ABA-treated *cop1-4* mutants (Fig. 5a, b). On the other hand, unlike WT, dark treatment could not induce stomatal closure of *cop1-4* mutants, and subsequent FC treatment also did not open the stomata. These results indicate that stomata of *cop1-4* mutants are insensitive to dark-induced stomatal closure and FC-induced stomatal opening, and blocking PM H$^+$-ATPase activity is sufficient to restore constitutive open stomata phenotype of *cop1-4* mutants.

PM H$^+$-ATPases pump intracellular proton out of the cell, thereby generating electrochemical proton gradient across PM and mediating various physiological responses such as salt and alkali tolerances[53,54]. Enhanced PM H$^+$-ATPase activity of *cop1-4* had prompted us to investigate alkali tolerance of *cop1-4* mutants, as salt tolerance of *cop1-4* mutants was described previously[55]. We generated MS media with pH of 3.6, 5.8, and 8.0, and then transferred 11-day-old WT and *cop1-4* seedlings. After grown vertically for 7 days, we observed that both WT and *cop1-4* mutants grown in pH 8.0 exhibited reduced fresh weight compared to pH 5.8, but this decrease was smaller in *cop1-4* mutants (Fig. 5c, d). Meanwhile, primary root length of WT and *cop1-4* mutants grown in pH 8.0 showed no significant differences. These data suggest that enhanced PM H$^+$-ATPase activity confers alkaline tolerance to *cop1-4* mutants, especially resistance to alkalinity-driven growth inhibition.

## Discussion
Arabidopsis COP1 is a central light signaling repressor that suppresses stomatal opening in guard cells by inhibiting light-induced opening and promoting ABA-induced closure (Fig. 6). Our study identified a new function of COP1 in pH regulation that converges light and ABA signaling pathways to control stomatal movement. We found that constitutive stomatal opening in *cop1* mutants was associated with significant alkalinization of vacuoles and cytosol, coupled with acidification of the apoplast, indicating altered basal pH levels in guard cells (Figs. 1, 2, and 3). This observation aligns with previous research showing that pH changes inside the vacuole and cytosol typically occur in the same direction[46]. It is intriguing that mutations in genes other than pumps or channels can affect pH levels. For instance, mutations in the TMK1, a gene involved in auxin signaling, led to higher apoplastic pH levels due to its direct phosphorylation and activation of PM H$^+$-ATPases[27,56]. In our study, we demonstrated that COP1 negatively regulates the activity of PM H$^+$-ATPases (Fig. 4), shedding light on the mechanism by which COP1 controls intracellular pH and ultimately governs stomatal movement and plant growth as to light.

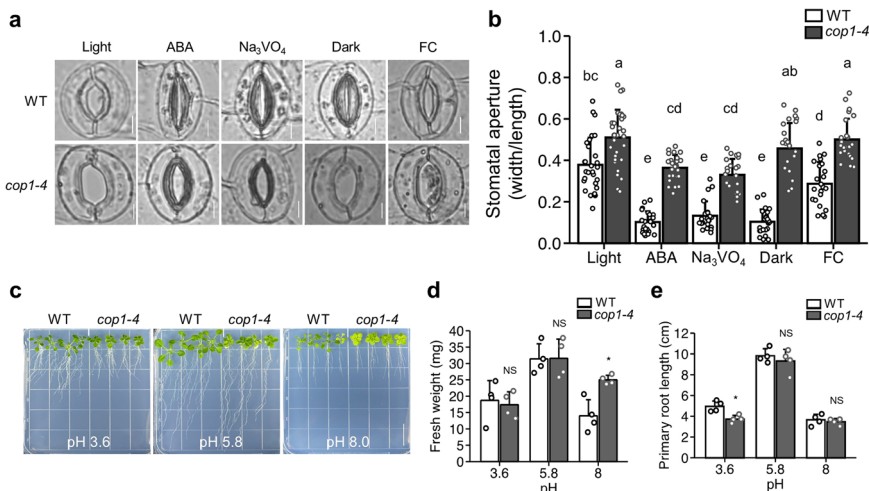

**Fig. 5 Enhanced plasma membrane (PM) H⁺-ATPase activity of *cop1-4* mutants affects plant growth and stomatal behavior. a** Detached leaves of 11-day-old wild-type (WT) and *cop1-4* mutant seedlings were incubated with stomatal opening buffer for 2 h, and then treated with various conditions as Fig. 2, except for abscisic acid (ABA) treated for 1 h. Guard cells were incubated with 1 mM $Na_3VO_4$ to investigate the role of H⁺-ATPase on stomatal aperture. (Scale bar, 5 μm.) **b** Quantification of stomatal apertures of guard cells from (**a**). Stomatal aperture index calculated by dividing the stomatal width by its length. Different letters indicate statistically significant differences between the plants ($P < 0.05$); one-way ANOVA; Tukey's HSD. **c** 11-day-old seedlings of WT and *cop1-4* mutants were transferred to ½ MS media at varying pH levels (3.6, 5.8, and 8.0) and grown vertically. Photographs and physiological assays were conducted 7 days after transferring. (Scale bar, 2 cm.) **d, e** Corresponding quantification of fresh weight (**d**) and primary root length (**e**) of plants from (**c**). *$P < 0.05$ and NS, not significant ($P > 0.05$); two-sample $t$ test; $n = 4$ each.

The constitutive opening of stomata with larger apertures in *cop1* mutants indicates that COP1 plays a negative role in stomatal opening. This phenotype was further characterized by expanded vacuoles with higher alkalinity compared to WT guard cells (Fig. 1). Two disrupted pH signaling mechanisms are attributable to the impaired stomatal closure of *cop1* mutants upon ABA treatment: the highly alkalinized cytosol and acidified apoplast diminish the effectiveness of conveying external signals into the vacuoles (Fig. 2, 3), and the vacuoles fail to undergo acidification for volume reduction (Fig. 1)[32,37,49]. Our experiments showed that guard cells around ABA-treated closed stomata display vacuolar acidification (Fig. 1), cytosolic acidification (Supplementary Fig. 4), and apoplastic alkalinization (Fig. 3). In the case of *cop1-4* mutants, they maintained alkaline vacuole (Fig. 1) and cytosol (Fig. 2 and Supplementary Fig. 4), and acidified apoplast (Fig. 3), which showed the opposite pH profiles with closed stomata.

The increase of cytosolic pH in *cop1-4* mutants is consistent with the phenotype of *slac1-1* mutants, which are also defective in ABA-induced stomatal closure due to impaired anion loss from the cytosol[6]. Moreover, *cop1-4* mutants show a minor increase in apoplastic pH compared to WT with ABA treatment, which hinders the increase in ABA concentration under stress conditions. Normally, the apoplast alkalinizes under stress conditions, which helps increase the ABA concentration[49]. The increased intracellular pH and decreased intercellular pH observed in *cop1-4* mutants could either be a cause or a consequence of their constantly photomorphogenic traits, and it is similar to the pH profile observed in photosynthesis-activating cells in response to light exposure in plants[46,57].

In this study, we observed stable cytosolic pH of *cop1-4* mutants upon ABA, dark, and FC treatment (Fig. 2). It was previously reported that 10 min of ABA treatment induces cytosolic alkalinization[32,37], which pH dynamic is opposite to dark-induced cytosolic acidification (Fig. 2). Our experiment also resulted in similar cytosolic alkalinization after 10 min of ABA treatment (Fig. 2), but prolonged ABA treatment about 1 h was sufficient to close the WT stomata and resulted in cytosolic

acidification (Supplementary Fig. 4). A study that treated ABA for 1 day also reported diminished PM H⁺-ATPase activity[58]. Here, we speculate that at the early stage of ABA signaling (about 0–10 min after ABA treatment), PM H⁺-ATPases are activated to induce pH gradient across the PM[37], but after sufficient signal transduction (about 1 h after ABA treatment), PM H⁺-ATPase activity decreases and induces cytosolic acidification and apoplastic alkalinization.

In the case of *cop1-4* mutants, basal pH of vacuole and cytosol are higher, and that of apoplast is lower than WT. This profile is maintained even after ABA or dark treatment, and vacuolar morphology and intracellular pH were even unchanged upon those treatments. Based on our observations about disrupted pH profiles of *cop1-4* mutants, we suspected that acidic apoplast and alkaline cytosol may be due to activation of PM H⁺-ATPases, ion pumps that export cellular protons out of the cells. Previous studies showed that activating or overexpressing PM H⁺-ATPases induced stomatal opening[31,59], which support the relationship between COP1 and PM H⁺-ATPases. Our rhizosphere acidification assay (Fig. 4a and Supplementary Fig. 7) and direct PM H⁺-ATPase activity assay (Fig. 4b, c) suggested that COP1 inhibits the proton pumping activity of PM H⁺-ATPases, which are primarily responsible for pH regulation stimulated by blue light (Fig. 4). This mechanism confirms the existing knowledge of the downstream function of COP1, which represses the PHOT-mediated blue light signaling pathway. It is worth noting that COP1 does not completely block the blue light signaling pathway, but rather regulates it quantitatively (Fig. 4). Supporting this is the observation that, cotyledon expansion is gradually reduced with an increase in COP1 amount under blue light[50]. The antagonistic relationship between COP1 and H⁺-ATPases is further embodied in a phenotypic way. Increased proton extrusion due to the hyperactivation of H⁺-ATPases results in enlarged stomata and lower leaf temperatures[17]. These effects are similar to those seen in *ost2-2D* and AHA1-overexpressing plants, reinforcing the role of COP1 in regulating H⁺-ATPases and affecting pH[31,59]. While COP1 is conventionally known for its activities in the nucleus, recent evidence shows that a significant

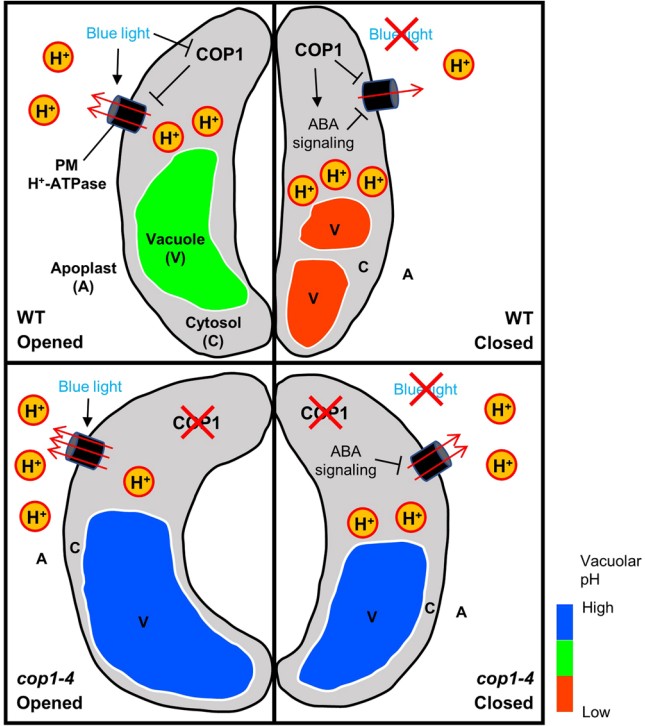

**Fig. 6 A hypothetical model for the role of COP1 in regulating stomatal movement through pH regulation.** In an opened wild-type (WT) guard cell (upper left panel), proton pumping by plasma membrane (PM) H$^+$-ATPase is downregulated by COP1. After stomata-closing signals such as ABA or dark (upper right panel), COP1 promotes ABA signaling, thereby inactivating PM H$^+$-ATPase activity again. Thus, cytosolic acidification and apoplastic alkalinization occur, and subsequent signal transduction induces vacuolar acidification and fragmentation, and finally stomatal closure. In an opened guard cell of cop1-4 mutant (lower left panel), absence of COP1 is responsible for enhanced PM H$^+$-ATPase activity. This elevated activity results in acidic apoplast and alkaline cytosol compared to WT. Stronger pH gradient across the PM induces signaling for stomatal opening, followed by enlarged alkaline vacuole and constitutive open stomata phenotype of cop1-4 mutants. After ABA or dark-mediated stomatal closure signals (lower right panel), only mild inhibitory effect can be posed to PM H$^+$-ATPase due to absence of COP1, a positive regulator of ABA signaling. Therefore, cop1-4 mutant's apoplastic pH is lower, and cytosolic pH remains higher than WT guard cells with closed stomata. Still, pH gradient across the PM is sufficient to induce stomatal opening, and the vacuole morphology and pH remain unaffected. Black arrows indicate stimulation, and blocked lines indicate inhibition. Red arrows indicate proton (H$^+$) movement. Relative pH of cytosol and apoplast are proportional to the numbers of "H$^+$"s inside respective regions. For graphical visibility, vacuoles at the upper side of the guard cells are omitted.

portion of COP1 is localized outside of the nucleus[60], and its activity in the cytosol has also been uncovered[61]. One possible molecular mechanism for the regulation of H$^+$-ATPases by COP1 involves ubiquitination of the H$^+$-ATPase protein through its E3 ligase activity, similar to yeasts[62]. Alternatively, COP1 may target 14-3-3 proteins, which are crucial for H$^+$-ATPase activation, to interrupt forming a stable complex with H$^+$-ATPase[63]. COP1's role in ABA signaling may be another possibility. COP1, as an E3 ubiquitin ligase, positively regulates ABA signaling via ubiquitination and degradation of clade A type 2 C phosphatases (PP2C) ABI/HAB and AHG3[17]. ABA signaling pathway also participates in regulating PM H$^+$-ATPase activity. ABA induces BRI1-ASSOCIATED RECEPTOR KINASE 1 (BAK1) phosphorylation and subsequent activation of AHA2[32], or inhibiting

Protein Phosphatase 1 (PP1) in *Vicia faba*, which mediates phosphorylation and activation of PM H$^+$-ATPases[64]. Further studies regarding direct connection between COP1, ABA signaling components and PM H$^+$-ATPases should be addressed.

Application of PM H$^+$-ATPase inhibitor to the guard cells efficiently induced stomatal closure of both WT and cop1-4 mutants, to the similar extent with ABA, which implies PM H$^+$-ATPases' significant role in stomatal movement (Fig. 5a, b). During stomatal opening process activated by blue light, photoreceptors mediate phosphorylation and activation of PM H$^+$-ATPases in guard cells[26,65,66]. Activated PM H$^+$-ATPases pump the protons out of the cells, thereby generating electrochemical gradient across the PM. This gradient induces the uptake of cations such as K$^+$, followed by anion accumulation in the cytosol[67]. Higher ion concentration results in increased water entry and turgor pressure, finally opening the stomata. To our experiments, cop1-4 mutants, with enhanced PM H$^+$-ATPase activity, maintain strong pH gradient (pH$_{cyt}$ − pH$_{apo}$) of about 1.6 pH unit, while that of WT was about 1.3 unit (Figs. 2, 3). Established pH gradient across the PM may contribute to enhanced K$^+$ uptake by potassium channels[68], upregulating stomatal opening process mentioned above. Acidic apoplast can also attribute to open stomata phenotype of cop1-4 mutants. Acid-growth theory states that proton extrusion-induced apoplastic acidification promotes cell expansion via activating cell wall loosening enzymes, which can make cells bigger through their turgor pressure[69,70]. Even under ABA treatment, guard cells of cop1-4 mutants exhibited more acidic apoplast than WT, and vacuolar morphology and pH were unchanged, which explain cop1-4 mutants' insensitivity towards ABA (Figs. 2, 3). ABA treatment and Na$_3$VO$_4$ treatment could not induce full closure of cop1-4 stomata, and we carefully speculate that this is due to vacuole inflexibility of cop1-4 mutants. The specific mechanism of how COP1 regulates vacuolar morphology and pH is largely unknown yet. One possible mechanism will be that COP1 can directly regulate tonoplast-residing ion pumps such as V-ATPases or V-PPases, which participate in guard cell signal transduction[67]. Other possibility may be that modified pH gradient across the tonoplast because of highly alkalinized cytosol can contribute to impaired signal transduction or enzymatic activity of tonoplast proteins[71]. Na$_3$VO$_4$, which is a phosphate analog, inhibits the activities of enzymes requiring phosphate compounds, such as P-type ATPases including PM H$^+$-ATPases and protein tyrosine phosphatases (PTPs)[72,73]. As Arabidopsis has several P-type ATPases and PTPs mediating ion transport and signal transduction, inhibition of those proteins with Na$_3$VO$_4$ may affected our in vivo experiments, resulting in incomplete stomatal closure of cop1-4 mutants. Indeed, previous study showed that inhibiting the activity of PTPs prevented stomatal closure after various stomata-closing cues[73]. Still, combined with our results (Figs. 1–4) and previous studies emphasizing important roles of PM H$^+$-ATPases on stomatal responses, we assume that elevated activity of PM H$^+$-ATPases is responsible for abnormal stomatal responses of cop1-4 mutants, even side effects of Na$_3$VO$_4$ may exist.

Owing to their high PM H$^+$-ATPase activity, cop1-4 mutants display tolerance to root-induced alkali stress in the context of fresh weight (Fig. 5c–e) Previously, it was reported that SOS3-LIKE CALCIUM BINDING PROTEIN 3 (SCaBP3)/CALCINEURIN B-LIKE 7 (CBL7) directly binds to AHA2 and promote autoinhibition of H$^+$-ATPase activity[54]. After alkali stimuli, SCaBP3/CBL7 is disassociated with AHA2 and subsequent increased H$^+$-ATPase activity enhances alkali tolerance. Indeed, mutation of SCaBP3/CBL7 caused increase in PM H$^+$-ATPase activity and alkali tolerance[54]. Moreover, root-specific activation of OST2/AHA1 resulted in plant growth and nutrient

accumulation in shoot[74]. COP1 may also participate in these processes by negatively regulating PM H$^+$-ATPase activity, mediating pH dynamics of cells and physiological responses under various stress conditions.

Regulating intracellular pH is vital for eukaryotic cells because it affects enzyme activities and protein modification, and maintaining it within physiological ranges is essential[75]. To cope with volatile environments, plants, as sessile organisms, have evolved to acquire vacuoles that provide greater buffering capacity, and have conserved two major proton pumps, PM H$^+$-ATPases and V-PPase, across plant taxa[76]. However, dysregulated pH in cop1-4 mutants with constitutively increased cytosolic pH and decreased apoplastic pH is reminiscent of cancer cells that capitalize on an alkaline intracellular pH for cell proliferation and apoptosis evasion while consuming more glucose[77]. Similarly, cop1 mutants exhibit higher biomass and increased sucrose uptake[55], finding parallels in cancer cells with respect to constitutive growth and elevated pH gradient. Thus, the pH gradient, as a ubiquitous energy source, is critical for cellular activities across organisms for biological growth and maintenance. This study provides physiological evidence for the role of COP1 in pH regulation to modulate plant growth and offers a perspective on the intricate mechanisms of pH-dependent cellular behavior.

## Materials and methods

**Plant materials and growth conditions**. All mutants and transgenic *A. thaliana* plants were from the Columbia-0 (Col-0) ecotype background and *cop1-4* were used in this study[78]. Col-0 was used as WT for all experiments. T3 homozygous plants of the *pUBQ10::ClopHensor* transgenic lines (Col-0 and *cop1-4* background)[35], were used for assays.

Seeds were surface-sterilized with 70% ethanol, sown on half-strength Murashige and Skoog (1/2 MS) medium containing 1% sucrose and 0.6% plant agar (pH 5.8), stratified in the dark at 4 °C for 3 days, and then grown under long-day conditions (16 h light/ 8 h dark) at 22 °C and 60% relative humidity in growth chambers. Unless otherwise stated, 8-day-old seedlings were transferred to and grown on soil as one plant per pot (6 cm in length and height) under short-day conditions (8 h light/16 h dark) at 22 °C and 60% relative humidity in environmentally controlled growth chambers. Four-week-old plants were used for experiments, unless otherwise noted.

For the phenotypic analysis of plants grown on media with varying pH, 11-day-old plants grown horizontally were transferred to half-strength MS medium with the varying pH levels (3.6, 5.8, 8.0) that were adjusted with 2.5 mM MES-BTP, except for the pH 3.6 medium. To prepare the pH 3.6 medium, agar-containing medium was later mixed with MS medium after autoclaving due to agar hydrolysis at low pH. After transferring the seedlings to the manipulated pH medium, they were grown vertically for an additional 7 days. All physiological results were obtained with 18-day-old plants.

**Imaging and measuring cellular pH**. Imaging and measuring cellular pH were conducted using the Leica SP8 X confocal laser scanning microscope equipped with a HC PL APO CS2 x40 water immersion objective.

To measure vacuolar pH in guard cells, the fluorescent cell-permeant dye BCECF-AM (Molecular Probes) was used with slight modifications[36,42]. Specifically, dye loading was achieved by submerging the epidermal peels in stomatal opening buffer (5 mM MES-BTP (pH 6.5), 50 mM KCl, 100 μM CaCl$_2$) with 10 μM BCECF-AM and 0.02% Pluronic F-127 (Invitrogen) to facilitate dye loading, followed by 90 min of staining at 22 °C in darkness. After staining, the epidermal peels were washed once in

dye-free buffer. The peels were then incubated in stomatal opening buffer for 2 h in light at 22 °C and either directly imaged or treated with 10 μM ABA for 1 h or darkness for 1 h. Fluorescence signals for protonated BCECF (excitation, 458 nm; emission, 514 nm) and deprotonated BCECF (excitation, 488 nm; emission, 514 nm) were detected. The ratio (488/458) was then converted into absolute pH values based on the calibration curve (Supplementary Fig. 1). In vivo calibration of BCECF was performed in guard cells, which were incubated for 30 min in pH equilibration buffers containing 50 mM citrate buffer-BTP (pH 4.3–5.0), 50 mM MES-BTP (pH 5.0-6.3) or 50 mM HEPES-BTP (pH 6.3–7.8) and 100 μM nigericin (Sigma-Aldrich). Ratio values were plotted against pH, and the calibration curves were generated from log-transformed polynomial regression using R software.

Guard cell cytosolic pH was measured through genetically-encoded pH sensors using plants expressing *pUBQ10::ClopHensor*[35]. Intact leaves were mounted on the slide glass and instantly used for imaging. Fluorescence signals for protonated one (excitation, 458 nm; emission, 514 nm) and deprotonated one (excitation, 488 nm; emission, 514 nm) were detected, and the relative cytosolic pH was calculated by 488/458. For in vivo calibration of ClopHensor, guard cells of WT expressing ClopHensor were incubated for 30 min in pH equilibrium buffers containing 50 mM HEPES-BTP (pH 6.3–7.8) and 50 mM ammonium acetate[35]. Ratio values were plotted against pH, and the calibration curves were generated from linear regression using R software.

Apoplastic pH measurement was performed using a ratiometric fluorescent dye HPTS[47]. Intact leaves were mounted on a block of solid stomatal opening buffer described above supplemented with 1 mM of 8-hydroxypyrene-1,3,6-trisulfonic acid trisodium salt (HPTS; Alfa Aesar) and instantly used for imaging. Fluorescence signals for the protonated HPTS (excitation, 405 nm; emission, 514 nm) and deprotonated HPTS (excitation, 488 nm; emission, 514 nm) were detected. In situ calibration of HPTS was performed with pH equilibration buffers used for in vivo BCEFC calibration containing 1 mM of HPTS. Ratio values were plotted against pH, and the calibration curves were generated from log-transformed linear regression.

**pH imaging analysis**. Fiji software was used to analyze the images with batch processing from a custom ImageJ macro. Manual thresholding was applied to mask and remove noise signals. To calculate the relative pH, the pixel value of the deprotonated signal was divided by that of the protonated signal. The ratio values were represented using a color lookup table, with the same minimum and maximum values set for each experiment, NA values for white color. The relative pH values were transformed into absolute pH values using the equation derived from the calibration curve. Calculated values were averaged over a pair of guard cells and statistically evaluated in R.

**Vacuolar analysis**. Guard cell vacuolar morphology was assessed using the *pVHP1::VHP1-sGFP* transgenic lines expressing the tonoplast marker[41]. GFP fluorescence was excited at 488 nm with an emission band of 500–500 nm. To estimate vacuolar volume, the BCECF dye was used as a vacuolar loading dye, and vacuolar occupancy was calculated as the ratio of vacuolar volume to the guard cell volume.

**Rhizosphere acidification assay**. 11-day-old plants, initially grown horizontally, were transferred to half-strength Murashige and Skoog (MS) medium (1 mM MES-KOH, pH 6.3), supplemented with 0.003% bromocresol purple, 1% sucrose, and 0.8% plant agar, and grown vertically for 36 h in growth chamber with

the noted condition. In the case of additional treatments, a 4 mM $Na_3VO_4$ medium was prepared using a 200 mM sodium orthovanadate (Sigma-Aldrich) stock solution (pH 9.4), while a 10 µM abscisic acid (ABA) medium was from a 10 mM ABA stock. No further pH adjustment was undertaken while preparing the $Na_3VO_4$ medium using KOH. The plates were visually examined and the color was evaluated.

**Plasma membrane vesicle isolation**. Plants were grown in potting soil in a growth chamber under long-day conditions (16 h light/8 h dark) at 22 °C and 60% relative humidity. Three-week-old plant materials were prepared for isolating plasma membrane vesicles using the aqueous two-phase separation method[51]. Fresh plants were homogenized with an ice-cold homogenization buffer (0.33 M sucrose, 10% (v/v) glycerol, 0.2% BSA, 0.2% casein hydrolysate, 0.6% (w/v) polyvinylpyrrolidone, 5 mM ascorbic acid, 5 mM EDTA, 5 mM dithiothreitol (DTT), 1 mM phenylmethylsulfonyl fluoride (PMSF), 1x protease inhibitor and 50 mM HEPES-KOH, pH 7.5). The homogenate was filtered through two layers of Miracloth and centrifuged at 13,000 $g$ for 10 min. The supernatant was then centrifuged at 100,000 $g$ for 1 h to obtain a microsomal pellet that was resuspended in buffer I (0.33 M sucrose, 3 mM KCl, 1 mM DTT, 0.1 mM EDTA, 1× protease inhibitor, and 5 mM potassium phosphate, pH 7.8). The PM (upper phase) was obtained using pre-cold two-phase mixture (6.2% (w/w) Dextran T-500 and 6.2% (w/w) polyethylene glycol 3350 in 5 mM potassium phosphate (pH 7.8), 0.33 M sucrose, and 3 mM KCl) by centrifuging the tubes at 1000 $g$ for 5 min, repeating three times to obtain the milky-white and transparent upper phase. The final upper phases were collected and diluted with buffer II (0.33 M sucrose, 0.1 mM EDTA, 1 mM DTT, 1× protease inhibitor, 20 mM HEPES–KOH, pH 7.5), and then centrifuged at 100,000 $g$ for 1 h. The resulting pellet was collected and resuspended with buffer II containing 1 mM EDTA to obtain the PM.

**PM $H^+$-ATPase activity assay**. The isolated PM vesicles were used to measure the $H^+$-transport activity[51,52]. The $H^+$ produced by PM $H^+$-ATPase from inside-out vesicles formed an inside-acid pH gradient (pH) in the vesicles, and it was measured as a decrease (quench) in the fluorescence by a pH-sensitive fluorescence probe quinacrine (Sigma-Aldrich). Fifty micrograms of PM vesicles were incubated with 2 ml of reaction buffer (140 mM KCl, 3 mM $MgSO_4$, 5 µM quinacrine, 0.05% (w/v) Brij-58, and 20 mM HEPES-BTP, pH 7.0) for 5 min in the darkened chamber of a spectrofluorometer (FP-8500) with 430 nm excitation and 500 nm emission wavelength. After the fluorescence was stabilized, the reaction was initiated by adding 3 mM ATP (final concentration) and the fluorescence was recorded for 5 min. At the end of the reaction, 10 µM (final concentration) protonophore nigericin was added to the assay buffer to dissipate the pH gradient and stop the reaction.

**Examination of PM $H^+$-ATPase protein levels**. The isolated PM vesicle samples of WT and cop1-4 mutants were subjected to SDS-PAGE. Following 8% SDS-PAGE, PM $H^+$-ATPase protein levels were examined by western blotting with anti-PM $H^+$-ATPase antibody (AS07-260; Agrisera). The same PVDF membrane used in the western blotting was stained with Coomassie Brilliant Blue, which was used as a loading control.

**Quantitative RT-PCR**. Total RNA was extracted from 14-d-old seedlings using the Plant Total RNA Purification Mini kit (Favorgen) according to the manufacturer's protocol. RNA was then diluted to 10 ng/µl and reverse-transcribed into cDNA using the ReverTra Ace qPCR RT Master Mix with gDNA Remover

(TOYOBO). qRT-PCR was performed using the TOPreal qPCR PreMIX kit (Enzynomics) in a 20 µl reaction on a Real-Time PCR LightCycler 480 System (Roche). Gene-specific primers were designed for AHA1 (forward: 5′-GGACAAGTTCGGTGTG AGGT3′, reverse: 5′-ACCAACTCCTTGACCTGGTG-3′) and AHA2 (forward: 5′-TGCTCAAAGGACACTTCACG-3′, reverse: 5′-GCCCTTTAGCTTCACGACTG-3′). Relative expression values were determined using UBQ10 (forward: 5′-GGCCT TGTATAATCCCTGATGAATAAG-3′, reverse: 5′-AAAGAGAT AACAGGAACGGAAACATAGT-3′) as an internal reference and the comparative Ct method ($2^{-\Delta\Delta Ct}$). All qPCR analyses were performed in triplicate using three independent biological replicates.

**Stomatal aperture analysis**. Detached leaves of Arabidopsis seedlings were treated with various buffers as indicated. Epidermal peels were mounted on a slide glass, and photographs of the abaxial leaf surface were captured using a Nikon Eclipse i80 microscope equipped with a camera (Nikon). The width and length of the stomatal aperture were measured using Fiji software, and the stomatal aperture index was calculated by dividing the aperture width by the length[79].

**Statistics and reproducibility**. All statistical analyses were conducted using R version 4.0.3[80] and Microsoft Excel. The analysis included a two-sample t-test, Wilcoxon rank-sum test, and one-way or two-way analysis of variance (ANOVA) with Tukey's HSD. Each experiment was performed at least three times.

**Reporting summary**. Further information on research design is available in the Nature Portfolio Reporting Summary linked to this article.

## Data availability
All data presented in this study are included in the main text and Supplementary Information. All source data of the main figures are included in the Supplementary Data 1. Uncropped blot images of Supplementary Fig. 8 are shown in Supplementary Fig. 9. Any additional data and plants are available upon reasonable request.

## Code availability
The scripts used for the pH analysis (dual excitation ratio values and its calibration to the absolute pH) are available at https://github.com/SeoyeonCha/pH_analysis.

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

## Acknowledgements
This work was supported by the National Research Foundation of Korea (NRF) grant funded by the Korean government (MSIT) (Project No. 2021R1A2C1003446).

## Author contributions
H.S.S. designed research; S.C. and W.K.M. performed research; S.C., W.K.M., and H.S.S. analyzed data; and S.C., W.K.M., and H.S.S. wrote the paper.

## Competing interests
The authors declare no competing interests.
