## [Peer Review File · Communications Biology]

Reviewers' comments:

Reviewer #1 (Remarks to the Author):

The involvement of COP1, a light signal component, in the regulation of stomatal movement and development is well-known. However, the precise regulatory mechanism of COP1 in these physiological and developmental processes remains unclear, particularly with regards to the constitutive stomatal opening caused by COP1 functional defects. While some studies have suggested that COP1 regulates stomatal movement through the ABA signaling pathway, its role as a light signaling component in light-induced stomatal movement is unknown. This paper presents an analysis of the effects of COP1 on the pH of the cytoplasm, vacuole, and apoplastic of guard cells, and reveals a new mechanism by which COP1 regulates stomatal movement through the regulation of proton pumping activity. This finding provides a new basis for understanding the constitutive stomatal opening caused by the loss of COP1 function.

Two issues should be noted in this study. Firstly, in the experiment on rhizosphere acidification, it is recommended to include a control group treated with Na₃VO₄ to better demonstrate that rhizosphere acidification is caused by proton pump action rather than the secretion of acidic substances by the root system. Secondly, it is recommended to conduct an analysis of root H⁺-ATPase activity to more rigorously explain how COP1 negatively regulates proton pumping activity. In conclusion, a single result on rhizosphere acidification is insufficient to demonstrate that COP1 negatively regulates proton pump activity.

Reviewer #2 (Remarks to the Author):

Short summary:

In this manuscript, authors claim that COP1 affects inter-/intra-cellular pH, thereby modulating stomatal movement. COP1 is a master negative regulator of photomorphogenesis, mediating pleiotropic cellular responses in plants. COP1 is known to repress stomatal opening, and several reports have shown its involvement of microtubule destabilization, PP2C-A degradation or subsequent activation of slow-type anion channels in guard cells. However, the direct mechanistic link through which COP1 modulates stomatal aperture remains to be elucidated. Based on the literatures reporting altered cellular pH accompanying stomatal movements, the authors have examined whether COP1 function is also associated with pH homeostasis. Although the finding of altered cellular pH in *cop1* mutant is interesting, there are the following serious concerns about this manuscript as a publication in Communications Biology.

Main shortcoming:

The altered cellular pH in *cop1* mutant does not fully elucidate the role of pH as a "secondary messenger" (line 19) in stomatal movement. This is because, as described in the Introduction, "Cytosolic alkalization has been implicated in both ABA-induced stomatal closure and fusicoccin (FC)-induced stomatal opening, making it difficult to uncover its homeostatic mechanism" (lines 61-64).

The data shown in Fig. 5, which may support the authors' hypothesis, are not entirely convincing to me. They claim that artificial alkalization of extracellular pH rescued the stomatal and growth phenotypes in *cop1*; however, they only altered the pH of the growth medium on which the roots were placed. It is not clear whether, how, and to what extent the pH in the guard cells is affected under such conditions, considering the presence of pH homeostatic mechanisms in the whole plant. This aspect should be addressed at least.

Methodological comments:

Fig. 1. The calibration for the vacuolar pH measurement may not be accurate, as only the pH of the

incubation buffer for the epidermal strips was altered, without guaranteeing that the vacuolar pH is the same as the buffer. To improve the accuracy of the calibration, I strongly suggest utilizing a buffer with an uncoupler such as nigericin (see Ref. 1), or conducting *in vitro* calibration (see Ref. 2).

Fig. 2. It is not clear what the light condition was during the experiments. I suggest examining the cytosolic pH "dynamics" in response to light or ABA treatment, as done in Fig. 1 and 3, and discussing proton transfer across the plasma membrane or tonoplast in *cop1*.

Fig. 5. Although the bromocresol purple experiment suggested higher proton extrusion in *cop1*, it does not necessarily reflect the activity of the PM H⁺-ATPase alone. Other proton-transferring mechanisms, such as proton symporters, antiporters, or proton leakage across the membrane, could also be affected in *cop1*. To support their model (Fig. 6), direct measurements of PM H⁺-ATPase activity in *cop1* *in vivo* or *in vitro* (see Ref. 3), are required.

Ref. 1: Phosphorylation of the plasma membrane H⁺-ATPase AHA2 by BAK1 is required for ABA-induced stomatal closure in Arabidopsis., Pei D, Hua D, Deng J, Wang Z, Song C, Wang Y, Wang Y, Qi J, Kollist H, Yang S, Guo Y, Gong Z. Plant Cell. 2022 Jul 4;34(7):2708-2729. doi: 10.1093/plcell/koac106.

Ref. 2: Evaluation of BCECF fluorescence ratio imaging to properly measure gastric intramucosal pH variations *in vivo*., Rochon P, Jourdain M, Mangalaboyi J, Fourrier F, Soulié-Bégu S, Buys B, Dehlin G, Lesage JC, Chambrin MC, Mordon S., J Biomed Opt. 2007 Nov-Dec;12(6):064014. doi: 10.1117/1.2821698.

Ref. 3: Blue light activates the plasma membrane H⁽⁺⁾-ATPase by phosphorylation of the C-terminus in stomatal guard cells. Kinoshita T, Shimazaki Ki. EMBO J. 1999 Oct 15;18(20):5548-58. doi: 10.1093/emboj/18.20.5548.

Reviewer #3 (Remarks to the Author):

Despite the well-established role of COP1 in guard cell signaling, the underlying cellular mechanisms by which it affects guard cell responses to any environmental and endogenous signals are still unclear. In this manuscript, Cha et al. aim to reassess how COP1 modulates stomatal movement, with a particular emphasis on cellular pH as a key biochemical factor for stomatal regulation. Here, they show that mutations in COP1 led to larger stomatal apertures, distinct vacuolar dynamics and disrupted pH dynamics within guard cells. Specifically, *cop1* mutants bore increased pH in vacuoles and cytosol and reduced apoplastic pH, leading to an inability in normal stomatal responses to ABA and dark. The observed changes in cellular pH were attributed to the increased proton efflux activity of *cop1* mutants. Furthermore, they discovered that manipulating external pH conditions could rescue the altered stomatal physiology and plant growth in *cop1* mutant. Overall, this study highlights the crucial role of COP1 in maintaining pH homeostasis and the importance of guard cell pH in integrating various signals to determine stomatal movement.

The manuscript is well presented and easy to follow. However, I think there are several minor points that should be improved:

1.Line 72, "*cop1*" should be italicized.

2.Fig.1: According to the order of the results stated in the text, I suggest adjusting Fig. 1a and b to behind of Fig. 1c, d, e, and f.

3.Fig.5: "p" should be italicized.

4.Line 383, "*A. thaliana*" should be italicized.

Lines 422 and 431, Please add a space between the unit and the number, such as 2h and 50mM.

Response to Referees

(Summary of responses)

We appreciate all three reviewers for their comprehensive and insightful comments, which have significantly contributed to improving our original manuscript. Please note that we have addressed all concerns necessitating additional experiments or minor adjustments. We have highlighted these changes in underlined blue text for additions and strikethrough red text for deletions within the main text. Please find our detailed point-by-point responses provided in blue.

Reviewers' comments:

Reviewer #1 (Remarks to the Author):

The involvement of COP1, a light signal component, in the regulation of stomatal movement and development is well-known. However, the precise regulatory mechanism of COP1 in these physiological and developmental processes remains unclear, particularly with regards to the constitutive stomatal opening caused by COP1 functional defects. While some studies have suggested that COP1 regulates stomatal movement through the ABA signaling pathway, its role as a light signaling component in light-induced stomatal movement is unknown. This paper presents an analysis of the effects of COP1 on the pH of the cytoplasm, vacuole, and apoplastic of guard cells, and reveals a new mechanism by which COP1 regulates stomatal movement through the regulation of proton pumping activity. This finding provides a new basis for understanding the constitutive stomatal opening caused by the loss of COP1 function.

Two issues should be noted in this study. Firstly, in the experiment on rhizosphere acidification, it is recommended to include a **control group treated with Na₃VO₄** to better demonstrate that rhizosphere acidification is caused by proton pump action rather than the secretion of acidic substances by the root system. Secondly, it is recommended to conduct an analysis of **root H⁺-ATPase activity to more rigorously explain how COP1 negatively regulates proton pumping activity**. In conclusion, a single result on rhizosphere acidification is insufficient to demonstrate that COP1 negatively regulates proton pump activity.

(Response to reviewer)

Thank you for the comment.

Regarding the first concern, we have conducted additional experiments involving the treatment of the rhizosphere with Na_3VO_4 at concentrations of 4 mM. It was shown that Na_3VO_4 completely inhibited the rhizosphere acidification in both WT and *cop1-4* mutants, demonstrating that rhizosphere acidification is caused by the PM H^+ -ATPase activity, rather than the secretion of other acidic substances.

We have now included the results of the 4 mM Na_3VO_4 treatment in Supplementary Fig. 7, which serves as a control group to further elucidate the role of proton pump action in rhizosphere acidification. Additionally, we have described this result in the “Results” section (Revised manuscript page 8), and modified the “Materials and Methods” section (Revised manuscript page 19-20).

In response to the second concern, we conducted an analysis of PM H^+ -ATPase activity using PM vesicles isolated from rosette leaves of WT and *cop1-4* mutants and measured the pH gradient produced by PM H^+ -ATPases as a decrease in the fluorescence by a pH-sensitive fluorescence probe quinacrine.

As shown in Revised Fig. 4b and c, the results indicate that *cop1-4* mutants exhibit a substantial increase in PM H^+ -ATPase activity, approximately a 2.5-fold increase when compared to WT. PM H^+ -ATPase activity was inhibited upon Na_3VO_4 treatment (data shown below), further confirming the reliability of our experiments. We have described and discussed these results in the “Results” (Revised manuscript page 8) and “Discussion” (Revised manuscript page 12-13) sections, and detailed materials and methods for plasma membrane vesicle isolation and PM H^+ -ATPase activity assay are added to the “Materials and Methods” section (Revised manuscript page 20-21). We also added this result to the “Abstract” section (Manuscript page 2).

To determine whether this enhanced PM H⁺-ATPase activity was from an increase in protein levels or enhanced activity, we conducted immunoblotting using the same samples to detect PM H⁺-ATPase (AS07-260; Agrisera). Revised Supplementary Fig. 8 show that the protein levels of PM H⁺-ATPase were comparable between WT and *cop1-4* mutants, thereby confirming that the observed increase in PM H⁺-ATPase activity in *cop1-4* mutants was not due to an increase in protein quantity. We have added this result to the “Results” section (Revised manuscript page 8), and detailed materials and methods regarding this result to the “Materials and Methods” section (Revised manuscript page 21).

Although the precise mechanism behind this enhanced activity remains elusive, these results robustly support our proposed model regarding COP1's negative act on the proton pumping activity.

Reviewer #2 (Remarks to the Author):

Short summary:

In this manuscript, authors claim that COP1 affects inter-/intra-cellular pH, thereby modulating stomatal movement. COP1 is a master negative regulator of photomorphogenesis, mediating pleiotropic cellular responses in plants. COP1 is known to repress stomatal opening, and several reports have shown its involvement of microtubule destabilization, PP2C-A degradation or subsequent activation of slow-type anion channels in guard cells. However, the direct mechanistic

link through which COP1 modulates stomatal aperture remains to be elucidated. Based on the literatures reporting altered cellular pH accompanying stomatal movements, the authors have examined whether COP1 function is also associated with pH homeostasis. Although the finding of altered cellular pH in *cop1* mutant is interesting, there are the following serious concerns about this manuscript as a publication in Communications Biology.

Main shortcoming:

The altered cellular pH in *cop1* mutant does not fully elucidate the role of pH as a "secondary messenger" (line 19) in stomatal movement. This is because, as described in the Introduction, "Cytosolic alkalinization has been implicated in both ABA-induced stomatal closure and fusicoccin (FC)-induced stomatal opening, making it difficult to uncover its homeostatic mechanism" (lines 61-64).

(Response to reviewer)

Thank you for your comment.

We agree that subcellular pH may not act as a secondary messenger for stomatal movement, as both ABA and fusicoccin induce cytosolic alkalinization of guard cells, even they exert opposite effects to the stomatal aperture. To further examine the pH dynamics during stomatal movement, we assessed cytosolic pH after treating the guard cells with ABA for 1 hour, which condition that can fully induce stomatal closure in WT. The result shows that ABA treatment to the guard cells for 1 hour induced cytosolic acidification in both WT and *cop1-4* mutants (Revised Supplementary Fig. 4), while ABA treatment to the guard cells for 10 minutes caused cytosolic alkalinization in both WT and *cop1-4* mutants (Revised Fig. 2). These indicate that ABA transiently induce alkalinization of cytosol in guard cells, and after inducing stomatal closure, the cytosol stably maintains more acidic pH than normal condition. Besides, it seems obvious that *cop1-4* mutants have more alkaline vacuolar and cytosolic pH, and more acidic apoplastic pH than WT in stomatal opening/closing conditions (Revised Fig.1e, f, Revised Fig. 2, and Revised Fig. 3).

Thus, we carefully speculate that rather than secondary messenger, pH alterations during stomatal movement may be one of the factors for stomatal movement, which can reflect stomatal status.

Based on your comment and our additional experiments, we deleted the sentence "Overall, our study highlights the crucial role of COP1 in maintaining pH homeostasis and demonstrates how cellular pH can act as a secondary messenger in plants to coordinate growth and development." and added "Overall, our study highlights the crucial role of COP1 in maintaining pH homeostasis of guard cells by regulating PM

H⁺-ATPase activity, and demonstrates how proton movement affects stomatal movement and plant growth.”
(Revised manuscript page 2) We also changed the text “a secondary messenger” to “an important factor.”
(Revised manuscript page 4)

The data shown in Fig. 5, which may support the authors' hypothesis, are not entirely convincing to me. They claim that artificial alkalization of extracellular pH rescued the stomatal and growth phenotypes in *cop1*; however, they only altered the pH of the growth medium on which the roots were placed. It is not clear **whether, how, and to what extent the pH in the guard cells is affected under such conditions**, considering the presence of pH homeostatic mechanisms in the whole plant. This aspect should be addressed at least.

(Response to reviewer)

Thank you for your comment.

As you pointed out, there is no direct evidence that pH of the growth media can affect the pH of shoot part, especially the guard cells, of the seedlings placed on the media. In order to assess the impact of media pH on guard cell pH, WT expressing ClopHensor were grown in MS media with various pH profiles. Then we attempted to measure the cytosolic pH of each leaves from different pH media using confocal laser scanning microscope. Unfortunately, we failed to obtain convincing data supporting the correlation between media pH and guard cell pH, due to experimental limitations.

Besides, MS media used in Fig. 5 were titrated by adding KOH to the liquid MS media. This means that media with higher pH contained more K⁺, a cation that can induce stomatal opening. As increased K⁺ content in MS media can enhance K⁺ content in shoot part, which means those ions can be absorbed by roots and transported to shoot (ref. 1, ref. 2), we thought titration of the media using KOH might had additional effects to the stomatal aperture besides pH. Thus, we prepared MS media titrated with BIS-TRIS propane (BTP) to generate media with pH 3.6, 5.8, and 8.0, and transferred 11-day-old seedlings of WT and *cop1-4* mutants on those media. After 7 days, we observed fresh weight of *cop1-4* mutants placed on MS media with pH 8.0 was significantly higher than that of WT (Revised Fig. 5c, d), while primary root length was comparable with WT (Revised Fig. 5c, e). These data suggest that *cop1-4* mutants have tolerance to alkaline stress derived from root. We also assessed stomatal aperture of WT and *cop1-4* mutants, and found that the WT gained

larger stomatal aperture with increasing media pH (data shown below). In the case of *cop1-4* mutants, they exhibited similar stomatal aperture tendency with WT except for seedlings at pH 3.6, which had similar aperture with seedlings at pH 5.8 (data shown below). Still, at all conditions, *cop1-4* mutants showed larger stomatal aperture than WT. These data showing fresh weight and stomatal aperture under various media pH imply that media pH where roots were placed somehow influenced shoot part including guard cells, although specific mechanism remains unknown. We carefully speculate that enhanced PM H⁺-ATPase activity of *cop1-4* can partially rescue alkaline stress posed at roots, and that these results cannot explain how pH affects stomatal aperture. We have eliminated initial Fig. 5, and put the data described above to Revised Fig. 5. We also described and discussed the results in “Results” (Revised manuscript page 10) and “Discussion” sections. (Revised manuscript page 14-15) Detailed materials and methods are described in “Materials and Methods” section. (Revised manuscript page 17)

a Representative images showing stomatal apertures of plants used in (Fig. 5a) (Scale bar, 5 μm.)

b Quantification of stomatal apertures from (a). Stomatal aperture index was calculated by dividing the stomatal width by its length. Different letters indicate statistically significant differences between the plants ($P < 0.05$); one-way ANOVA; Tukey's HSD.

Focusing on the effects of enhanced PM H⁺-ATPase activity of *cop1-4* mutants on stomatal aperture, we treated WT and *cop1-4* mutants with PM H⁺-ATPase inhibitor sodium orthovanadate (Na₃VO₄) and compared the stomatal aperture. The results showed that Na₃VO₄ did closed the stomata of *cop1-4* mutants as well as WT, to similar extent with ABA (Revised Fig. 5a, b, note that stomatal aperture of *cop1-4* mutants treated with ABA and Na₃VO₄ are not distinguishable). This result strongly indicates that PM H⁺-ATPase activity is the key element of stomatal opening, and is responsible for enlarged stomatal pores of *cop1-4* mutants. We added the data to Revised Fig. 5, described and discussed at “Results” (Revised manuscript page 10) and “Discussion” sections. (Revised manuscript page 14-15)

Ref. 1: XQ Du, FL Wang, H Li, S Jing, M Yu, J Li, WH Wu, J Kudla, Y Wang. The Transcription Factor MYB59 Regulates K⁺/NO₃⁻ Translocation in the Arabidopsis Response to Low K⁺ Stress. *Plant Cell* **31**: 699-714. (2019) doi: 10.1105/tpc.18.00674

Ref. 2: FL Wang, YL Tan, L Wallrad, XQ Du, A Eickelkamp, ZF Wang, GF He, F Rehms, Z Li, JP Han, I Schmitz-Thom, WH Wu, J Kudla, Y Wang. A potassium-sensing niche in Arabidopsis roots orchestrates signaling and adaptation responses to maintain nutrient homeostasis. *Dev Cell* **56**: 781-794.e6. (2021) doi: 10.1016/j.devcel.2021.02.027

Methodological comments:

Fig. 1. The calibration for the vacuolar pH measurement may not be accurate, as only the pH of the incubation buffer for the epidermal strips was altered, without guaranteeing that the vacuolar pH is the same as the buffer. To improve the accuracy of the calibration, I strongly suggest utilizing a buffer with an uncoupler such as nigericin (see Ref. 1), or conducting in vitro calibration (see Ref. 2).

(Response to reviewer)

Thank you for your comment.

We changed the uncoupler from 50 mM ammonium acetate to 100 μ M nigericin in pH equilibrium buffer, and reperformed BCECF calibration. We obtained slightly decreased pH calibration data (0.02~0.2 pH unit), but still similar calibration curve (data shown below, Revised Supplementary Fig. 1). Based on these calibration data, we recalculated the vacuolar pH of the guard cells in Fig. 1 and Fig. S2 and changed the data (Revised Fig. 1e, f, and Revised Supplementary Fig. 2). We also revised the text “50mM ammonium acetate” in the “Materials and Methods” section to “100 μ M nigericin.” (Revised manuscript page 18)

However, in the case of *in vivo* ClopHensor calibration, we could not calibrate the pH using nigericin due to substantial decrease of the signal of genetically-encoded fluorescence ClopHensor. Instead, we employed pH equilibrium buffers containing 50 mM ammonium acetate with various pH, as previously described (ref. 3). We added the pH calibration data in Revised Supplementary Fig. 3, and detailed methods in “Materials and Methods” section. (Revised manuscript page 18)

Ref. 3: E. Demes, L Besse, P Cubero-Font, B Satiat-Jeuemaitre, S Thomine, A De Angeli. Dynamic measurement of cytosolic pH and $[\text{NO}_3^-]$ uncovers the role of the vacuolar transporter AtCLCa in cytosolic pH homeostasis. *Proc Natl Acad Sci U S A* **117**:15343-15353. (2020) doi: 10.1073/pnas.2007580117.

Fig. 2. It is not clear what the light condition was during the experiments. I suggest examining the cytosolic pH "dynamics" in response to light or ABA treatment, as done in Fig. 1 and 3, and discussing proton transfer across the plasma membrane or tonoplast in cop1.

(Response to reviewer)

Thank you for your comment.

Based on your insightful comment, we measured cytosolic pH using WT and *cop1-4* mutants each expressing ClopHensor treated with ABA, dark, or fusicoccin (Revised Fig. 2). When treated with ABA for 10 minutes, WT showed slight increase in cytosolic pH, which aligns with previous research (ref. 4), while *cop1-4* mutants maintained their cytosolic pH. Guard cells treated with dark condition showed cytosolic acidification in both WT and *cop1-4* mutants. In the case of fusicoccin treatment, cytosolic pH of both WT and *cop1-4* mutants increased. It seemed strange that ABA and dark condition, which both induce stomatal closure, change cytosolic pH in an opposite direction. However, after 1 hour of ABA treatment, which was sufficient to induce full closure of WT guard cells, both WT and *cop1-4* mutants exhibited cytosolic acidification (Revised Supplementary Fig. 4). These data indicate that when a stomatal pore is closed, the guard cells maintain more H⁺ inside their cytosol, while stomatal opening stimuli induce cytosolic alkalinization. Altogether with our vacuolar and apoplastic pH measurement, we suggest a model for COP1's role in stomatal movement by regulating pH dynamics of guard cells (Revised Fig. 6). We added these results in Revised Fig. 2, and explained them in "Results" section. (Revised manuscript page 6-7) Detailed explanation about pH dynamics in guard cells during stomatal movement is included in "Discussion" section. (Revised manuscript page 12, 14-15)

Ref. 4) D. Pei, D Hua, J Deng, Z Wang, C Song, Y Wang, Y Wang, J Qi, H Kollist, S Yang, Y Guo, Z Gong. Phosphorylation of the plasma membrane H⁺-ATPase AHA2 by BAK1 is required for ABA-induced stomatal closure in Arabidopsis. *Plant Cell* **34**: 2708-2729. (2022) doi: 10.1093/plcell/koac106

Fig. 5. Although the bromocresol purple experiment suggested higher proton extrusion in *cop1*, it does not necessarily reflect the activity of the PM H⁺-ATPase alone. Other proton-transferring mechanisms, such as proton symporters, antiporters, or proton leakage across the membrane, could

also be affected in *cop1*. To support their model (Fig. 6), direct measurements of PM H⁺-ATPase activity in *cop1* in vivo or in vitro (see Ref. 3), are required.

Ref. 1: Phosphorylation of the plasma membrane H⁺-ATPase AHA2 by BAK1 is required for ABA-induced stomatal closure in Arabidopsis., Pei D, Hua D, Deng J, Wang Z, Song C, Wang Y, Wang Y, Qi J, Kollist H, Yang S, Guo Y, Gong Z. Plant Cell. 2022 Jul 4;34(7):2708-2729. doi: 10.1093/plcell/koac106.

Ref. 2: Evaluation of BCECF fluorescence ratio imaging to properly measure gastric intramucosal pH variations in vivo., Rochon P, Jourdain M, Mangalaboyi J, Fourrier F, Soulié-Bégu S, Buys B, Dehlin G, Lesage JC, Chambrin MC, Mordon S., J Biomed Opt. 2007 Nov-Dec;12(6):064014. doi: 10.1117/1.2821698.

Ref. 3: Blue light activates the plasma membrane H⁽⁺⁾-ATPase by phosphorylation of the C-terminus in stomatal guard cells. Kinoshita T, Shimazaki Ki. EMBO J. 1999 Oct 15;18(20):5548-58. doi: 10.1093/emboj/18.20.5548.

(Response to reviewer)

Thank you for your comment.

Based on your valuable suggestions, we isolated PM vesicles from the rosette leaves of WT and *cop1-4* mutants, and subject them to PM H⁺-ATPase activity assay (ref. 5, ref. 6).

As shown in Revised Fig. 4b and c, the results indicate that *cop1-4* mutants exhibit a substantial increase in PM H⁺-ATPase activity, approximately a 2.5-fold increase when compared to WT. PM H⁺-ATPase activity was inhibited upon Na₃VO₄ treatment (data shown below), further confirming the reliability of our experiments. We have described and discussed these results in the “Results” (Revised manuscript page 8) and “Discussion” (Revised manuscript page 12-13) sections, and detailed materials and methods for plasma membrane vesicle isolation and PM H⁺-ATPase activity assay are added to the “Materials and Methods” section (Revised manuscript page 20-21). We also added this result to the “Abstract” section (Manuscript page 2).

To determine whether this enhanced PM H⁺-ATPase activity was from an increase in protein levels or enhanced activity, we conducted immunoblotting using the same samples to detect PM H⁺-ATPase (AS07-260; Agrisera). Revised Supplementary Fig. 8 show that the protein levels of PM H⁺-ATPase were comparable between WT and *cop1-4* mutants, thereby confirming that the observed increase in PM H⁺-ATPase activity in *cop1-4* mutants was not due to an increase in protein quantity. We have added this result to the “Results” section (Revised manuscript page 8), and detailed materials and methods regarding this result to the “Materials and Methods” section (Revised manuscript page 21).

Ref. 5: QS Qiu, Y Guo, MA Dietrich, KS Schumaker, JK Zhu. Regulation of SOS1, a plasma membrane Na⁺/H⁺ exchanger in *Arabidopsis thaliana*, by SOS2 and SOS3. *Proc Natl Acad Sci U S A* **99**. 8436-41 (2002) doi: 10.1073/pnas.122224699.

Ref. 6: Y Yang, Y Qin, C Xie, F Zhao, J Zhao, D Liu, S Chen, AT Fuglsang, MG Palmgren, KS Schumaker, XW Deng, Y Guo. The *Arabidopsis* chaperone J3 regulates the plasma membrane H⁺-ATPase through interaction with the PKS5 kinase. *Plant Cell* **22**, 1313-32 (2010) doi: 10.1105/tpc.109.069609.

Reviewer #3 (Remarks to the Author):

Despite the well-established role of COP1 in guard cell signaling, the underlying cellular mechanisms by which it affects guard cell responses to any environmental and endogenous signals are still unclear. In this manuscript, Cha et al. aim to reassess how COP1 modulates stomatal movement, with a particular emphasis on cellular pH as a key biochemical factor for stomatal regulation. Here, they show that mutations in COP1 led to larger stomatal apertures, distinct vacuolar dynamics and disrupted pH dynamics within guard cells. Specifically, *cop1* mutants bore increased pH in vacuoles and cytosol and reduced apoplastic pH, leading to an inability in normal stomatal responses to ABA and dark. The observed changes in cellular pH were attributed to the increased proton efflux activity of *cop1* mutants. Furthermore, they discovered that manipulating external pH conditions could rescue the altered stomatal physiology and plant growth in *cop1* mutant. Overall, this study highlights the crucial role of COP1 in maintaining pH homeostasis and the importance of guard cell pH in integrating various signals to determine stomatal movement.

The manuscript is well presented and easy to follow. However, I think there are several minor points that should be improved:

1. Line 72, “*cop1*” should be italicized.

(Response to reviewer)

We sincerely thank the reviewer for pointing out this typo.

We have now italicized “*cop1*” in the manuscript (Revised manuscript page 4).

2. Fig. 1: According to the order of the results stated in the text, I suggest adjusting Fig. 1a and b to behind of Fig. 1c, d, e, and f.

(Response to reviewer)

We appreciate the reviewer’s insightful suggestion to realign the figures as to the textual presentation of results.

Following your recommendation, we have repositioned Fig. 1a and b to appear after Fig. 1c, d, e, and f. Further, to enhance clarity regarding the dampened ABA response of *cop1* mutants, we have

also included vacuolar dynamics data for WT and *cop1-4* after the ABA treatment in the revised figure 1a, and described the result in the “Results” section. (Revised manuscript page 5) We then carefully revised the sentences in the manuscript to apply changed figure annotations.

3.Fig.5: “p” should be italicized.

(Response to reviewer)

Thank you for your comment.

We have now made the correction that italicizing "p" in Fig. 5.

4.Line 383, “*A. thaliana*” should be italicized.

(Response to reviewer)

Thank you for this finding.

We have now correctly italicized "*A. thaliana*". (Revised manuscript page 17)

Lines 422 and 431, Please add a space between the unit and the number, such as 2h and 50mM.

(Response to reviewer)

Thank you for bringing this to our attention.

We have added the required spaces between units and numbers, and we have also conducted a thorough review of the manuscript to ensure consistent formatting.

REVIEWERS' COMMENTS:

Reviewer #1 (Remarks to the Author):

The author has satisfactorily fulfilled all the requirements I specified and has addressed all the concerns that were raised.

Reviewer #2 (Remarks to the Author):

The authors have addressed my concerns well with the additional experiments and revised manuscript. I appreciate their effort to re-construct their model and conclusion.

Although I'm supportive for its publication in principle, I just have one suggestion: orthovanadate acts as a phosphate analogue, thereby inhibiting not only H⁺-ATPase but also other P-type ATPases and protein tyrosine phosphatases, etc. I suggest to note the possibility that the inhibition of these other targets might be involved in the in vivo effect.

Response to Referees

Reviewers' comments:

Reviewer #1 (Remarks to the Author):

The author has satisfactorily fulfilled all the requirements I specified and has addressed all the concerns that were raised.

(Response to reviewer)

We appreciate your valuable comments to our manuscript.

Reviewer #2 (Remarks to the Author):

The authors have addressed my concerns well with the additional experiments and revised manuscript. I appreciate their effort to re-construct their model and conclusion.

Although I'm supportive for its publication in principle, I just have one suggestion: orthovanadate acts as a phosphate analogue, thereby inhibiting not only H⁺-ATPase but also other P-type ATPases and protein tyrosine phosphatases, etc. I suggest to note the possibility that the inhibition of these other targets might be involved in the *in vivo* effect.

(Response to reviewer)

Thanks for your important comment. As you pointed out, orthovanadate can inhibit other kinds of enzymes that utilize phosphate compounds. This can cause unexpected side effects on *in vivo* experiments, including our experiments. Nevertheless, our pH measurements of guard cells (Fig. 1, 2, and 3) and PM H⁺-ATPase activity assay (Fig. 4b, c) give substantial clues supporting that COPI is involved in regulating PM H⁺-ATPase functions.

Fully agreeing with your suggestion, we added sentences explaining this possibility in the "Discussion" section stating 'Na₃VO₄, which is a phosphate analogue, inhibits the activities of

enzymes requiring phosphate compounds, such as P-type ATPases including PM H⁺-ATPases and protein tyrosine phosphatases (PTPs). As Arabidopsis has several P-type ATPases and PTPs mediating ion transport and signal transduction, inhibition of those proteins with Na₃VO₄ may affected our *in vivo* experiments, resulting in incomplete stomatal closure of *cop1-4* mutants. Indeed, previous study showed that inhibiting the activity of PTPs prevented stomatal closure after various stomata-closing cues. Still, combined with our results (Fig. 1-4) and previous studies emphasizing important roles of PM H⁺-ATPases on stomatal responses, we assume that elevated activity of PM H⁺-ATPases is responsible for abnormal stomatal responses of *cop1-4* mutants, even side effects of Na₃VO₄ may exist.' (Revised manuscript page 13-14).